# Video-to-Video Synthesis

**Ting-Chun Wang**[1], **Ming-Yu Liu**[1], **Jun-Yan Zhu**[2], **Guilin Liu**[1],
**Andrew Tao**[1], **Jan Kautz**[1], **Bryan Catanzaro**[1]
[1]NVIDIA, [2]MIT CSAIL
{tingchunw,mingyul,guilinl,atao,jkautz,bcatanzaro}@nvidia.com,
junyanz@mit.edu

## Abstract

We study the problem of video-to-video synthesis, whose goal is to learn a mapping
function from an input source video (e.g., a sequence of semantic segmentation
masks) to an output photorealistic video that precisely depicts the content of the
source video. While its image counterpart, the image-to-image translation problem,
is a popular topic, the video-to-video synthesis problem is less explored in the
literature. Without modeling temporal dynamics, directly applying existing image
synthesis approaches to an input video often results in temporally incoherent videos
of low visual quality. In this paper, we propose a video-to-video synthesis approach
under the generative adversarial learning framework. Through carefully-designed
generators and discriminators, coupled with a spatio-temporal adversarial objective,
we achieve high-resolution, photorealistic, temporally coherent video results on
a diverse set of input formats including segmentation masks, sketches, and poses.
Experiments on multiple benchmarks show the advantage of our method compared
to strong baselines. In particular, our model is capable of synthesizing 2K resolution
videos of street scenes up to 30 seconds long, which significantly advances the
state-of-the-art of video synthesis. Finally, we apply our method to future video
prediction, outperforming several competing systems. Code, models, and more
results are available at our website.

## 1 Introduction

The capability to model and recreate the dynamics of our visual world is essential to building
intelligent agents. Apart from purely scientific interests, learning to synthesize continuous visual
experiences has a wide range of applications in computer vision, robotics, and computer graphics.
For example, in model-based reinforcement learning [2, 24], a video synthesis model finds use in
approximating visual dynamics of the world for training the agent with less amount of real experience
data. Using a learned video synthesis model, one can generate realistic videos without explicitly
specifying scene geometry, materials, lighting, and dynamics, which would be cumbersome but
necessary when using a standard graphics rendering engine [35].

The video synthesis problem exists in various forms, including future video prediction [15, 18, 42, 45,
50, 64, 67, 70, 76] and unconditional video synthesis [59, 66, 68]. In this paper, we study a new form:
*video-to-video synthesis*. At the core, we aim to learn a mapping function that can convert an input
video to an output video. To the best of our knowledge, a general-purpose solution to video-to-video
synthesis has not yet been explored by prior work, although its image counterpart, the image-to-image
translation problem, is a popular research topic [6, 31, 33, 43, 44, 63, 65, 72, 81, 82]. Our method is
inspired by previous application-specific video synthesis methods [58, 60, 61, 74].

We cast the video-to-video synthesis problem as a distribution matching problem, where the goal is
to train a model such that the conditional distribution of the synthesized videos given input videos
resembles that of real videos. To this end, we learn a conditional generative adversarial model [20]

Figure 1: Generating a photorealistic video from an input segmentation map video on Cityscapes. Top left: input. Top right: `pix2pixHD`. Bottom left: `COVST`. Bottom right: `vid2vid` (ours). *The figure is best viewed with Acrobat Reader. Click the image to play the video clip.*

given paired input and output videos, With carefully-designed generators and discriminators, and a new spatio-temporal learning objective, our method can learn to synthesize high-resolution, photorealistic, temporally coherent videos. Moreover, we extend our method to multimodal video synthesis. Conditioning on the same input, our model can produce videos with diverse appearances.

We conduct extensive experiments on several datasets on the task of converting a sequence of segmentation masks to photorealistic videos. Both quantitative and qualitative results indicate that our synthesized footage looks more photorealistic than those from strong baselines. See Figure 1 for example. We further demonstrate that the proposed approach can generate photorealistic 2K resolution videos, up to 30 seconds long. Our method also grants users flexible high-level control over the video generation results. For example, a user can easily replace all the buildings with trees in a street view video. In addition, our method works for other input video formats such as face sketches and body poses, enabling many applications from face swapping to human motion transfer. Finally, we extend our approach to future prediction and show that our method can outperform existing systems. Please visit our website for code, models, and more results.

## 2 Related Work

**Generative Adversarial Networks (GANs).** We build our model on GANs [20]. During GAN training, a generator and a discriminator play a zero-sum game. The generator aims to produce realistic synthetic data so that the discriminator cannot differentiate between real and the synthesized data. In addition to noise distributions [14, 20, 55], various forms of data can be used as input to the generator, including images [33, 43, 81], categorical labels [52, 53], and textual descriptions [56, 79]. Such conditional models are called conditional GANs, and allow flexible control over the output of the model. Our method belongs to the category of conditional video generation with GANs. However, instead of predicting future videos conditioning on the current observed frames [41, 50, 68], our method synthesizes photorealistic videos conditioning on manipulable semantic representations, such as segmentation masks, sketches, and poses.

**Image-to-image translation** algorithms transfer an input image from one domain to a corresponding image in another domain. There exists a large body of work for this problem [6, 31, 33, 43, 44, 63, 65, 72, 81, 82]. Our approach is their video counterpart. In addition to ensuring that each video frame looks photorealistic, a video synthesis model also has to produce temporally coherent frames, which is a challenging task, especially for a long duration video.

**Unconditional video synthesis.** Recent work [59, 66, 68] extends the GAN framework for unconditional video synthesis, which learns a generator for converting a random vector to a video.

VGAN [68] uses a spatio-temporal convolutional network. TGAN [59] projects a latent code to a set of latent image codes and uses an image generator to convert those latent image codes to frames. MoCoGAN [66] disentangles the latent space to motion and content subspaces and uses a recurrent neural network to generate a sequence of motion codes. Due to the unconditional setting, these methods often produce low-resolution and short-length videos.

**Future video prediction.** Conditioning on the observed frames, video prediction models are trained to predict future frames [15, 18, 36, 41, 42, 45, 50, 64, 67, 70, 71, 76]. Many of these models are trained with image reconstruction losses, often producing blurry videos due to the classic regress-to-the-mean problem. Also, they fail to generate long duration videos even with adversarial training [42, 50]. The video-to-video synthesis problem is substantially different because it does not attempt to predict object motions or camera motions. Instead, our approach is conditional on an existing video and can produce high-resolution and long-length videos in a different domain.

**Video-to-video synthesis.** While video super-resolution [61, 62], video matting and blending [3, 12], and video inpainting [73] can be considered as special cases of the video-to-video synthesis problem, existing approaches rely on problem-specific constraints and designs. Hence, these methods cannot be easily applied to other applications. Video style transfer [10, 22, 28, 58], transferring the style of a reference painting to a natural scene video, is also related. In Section 4 , we show that our method outperforms a strong baseline that combines a recent video style transfer with a state-of-the-art image-to-image translation approach.

## 3   Video-to-Video Synthesis

Let $\mathbf{s}_1^T \equiv \{\mathbf{s}_1, \mathbf{s}_2, ..., \mathbf{s}_T\}$ be a sequence of source video frames. For example, it can be a sequence of semantic segmentation masks or edge maps. Let $\mathbf{x}_1^T \equiv \{\mathbf{x}_1, \mathbf{x}_2, ..., \mathbf{x}_T\}$ be the sequence of corresponding real video frames. The goal of video-to-video synthesis is to learn a mapping function that can convert $\mathbf{s}_1^T$ to a sequence of output video frames, $\tilde{\mathbf{x}}_1^T \equiv \{\tilde{\mathbf{x}}_1, \tilde{\mathbf{x}}_2, ..., \tilde{\mathbf{x}}_T\}$, so that the conditional distribution of $\tilde{\mathbf{x}}_1^T$ given $\mathbf{s}_1^T$ is identical to the conditional distribution of $\mathbf{x}_1^T$ given $\mathbf{s}_1^T$.

$$p(\tilde{\mathbf{x}}_1^T | \mathbf{s}_1^T) = p(\mathbf{x}_1^T | \mathbf{s}_1^T). \tag{1}$$

Through matching the conditional video distributions, the model learns to generate photorealistic, temporally coherent output sequences as if they were captured by a video camera.

We propose a conditional GAN framework for this conditional video distribution matching task. Let $G$ be a generator that maps an input source sequence to a corresponding output frame sequence: $\mathbf{x}_1^T = G(\mathbf{s}_1^T)$. We train the generator by solving the minimax optimization problem given by

$$\max_D \min_G E_{(\mathbf{x}_1^T, \mathbf{s}_1^T)}[\log D(\mathbf{x}_1^T, \mathbf{s}_1^T)] + E_{\mathbf{s}_1^T}[\log(1 - D(G(\mathbf{s}_1^T), \mathbf{s}_1^T))], \tag{2}$$

where $D$ is the discriminator. We note that as solving (2), we minimize the Jensen-Shannon divergence between $p(\tilde{\mathbf{x}}_1^T | \mathbf{s}_1^T)$ and $p(\mathbf{x}_1^T | \mathbf{s}_1^T)$ as shown by Goodfellow et al. [20].

Solving the minimax optimization problem in (2) is a well-known, challenging task. Careful designs of network architectures and objective functions are essential to achieve good performance as shown in the literature [14, 21, 30, 37, 49, 51, 55, 72, 79]. We follow the same spirit and propose new network designs and a spatio-temporal objective for video-to-video synthesis as detailed below.

**Sequential generator**. To simplify the video-to-video synthesis problem, we make a Markov assumption where we factorize the conditional distribution $p(\tilde{\mathbf{x}}_1^T | \mathbf{s}_1^T)$ to a product form given by

$$p(\tilde{\mathbf{x}}_1^T | \mathbf{s}_1^T) = \prod_{t=1}^T p(\tilde{\mathbf{x}}_t | \tilde{\mathbf{x}}_{t-L}^{t-1}, \mathbf{s}_{t-L}^t). \tag{3}$$

In other words, we assume the video frames can be generated sequentially, and the generation of the $t$-th frame $\tilde{\mathbf{x}}_t$ only depends on three factors: 1) current source frame $\mathbf{s}_t$, 2) past $L$ source frames $\mathbf{s}_{t-L}^{t-1}$, and 3) past $L$ generated frames $\tilde{\mathbf{x}}_{t-L}^{t-1}$. We train a feed-forward network $F$ to model the conditional distribution $p(\tilde{\mathbf{x}}_t | \tilde{\mathbf{x}}_{t-L}^{t-1}, \mathbf{s}_{t-L}^t)$ using $\tilde{\mathbf{x}}_t = F(\tilde{\mathbf{x}}_{t-L}^{t-1}, \mathbf{s}_{t-L}^t)$. We obtain the final output $\tilde{\mathbf{x}}_1^T$ by applying the function $F$ in a recursive manner. We found that a small $L$ (e.g., $L = 1$) causes training instability, while a large $L$ increases training time and GPU memory but with minimal quality improvement. In our experiments, we set $L = 2$.

Video signals contain a large amount of redundant information in consecutive frames. If the optical flow [46] between consecutive frames is known, we can estimate the next frame by warping the current frame [54, 69]. This estimation would be largely correct except for the occluded areas. Based on this observation, we model $F$ as

$$F(\tilde{\mathbf{x}}_{t-L}^{t-1}, \mathbf{s}_{t-L}^t) = (\mathbf{1} - \tilde{\mathbf{m}}_t) \odot \tilde{\mathbf{w}}_{t-1}(\tilde{\mathbf{x}}_{t-1}) + \tilde{\mathbf{m}}_t \odot \tilde{\mathbf{h}}_t, \tag{4}$$

where $\odot$ is the element-wise product operator and $\mathbf{1}$ is an image of all ones. The first part corresponds to pixels warped from the previous frame, while the second part hallucinates new pixels. The definitions of the other terms in Equation 4 are given below.

- $\tilde{\mathbf{w}}_{t-1} = W(\tilde{\mathbf{x}}_{t-L}^{t-1}, \mathbf{s}_{t-L}^t)$ is the estimated optical flow from $\tilde{\mathbf{x}}_{t-1}$ to $\tilde{\mathbf{x}}_t$, and $W$ is the optical flow prediction network. We estimate the optical flow using both input source images $\mathbf{s}_{t-L}^t$ and previously synthesized images $\tilde{\mathbf{x}}_{t-L}^{t-1}$. By $\tilde{\mathbf{w}}_{t-1}(\tilde{\mathbf{x}}_{t-1})$, we warp $\tilde{\mathbf{x}}_{t-1}$ based on $\tilde{\mathbf{w}}_{t-1}$.

- $\tilde{\mathbf{h}}_t = H(\tilde{\mathbf{x}}_{t-L}^{t-1}, \mathbf{s}_{t-L}^t)$ is the hallucinated image, synthesized directly by the generator $H$.

- $\tilde{\mathbf{m}}_t = M(\tilde{\mathbf{x}}_{t-L}^{t-1}, \mathbf{s}_{t-L}^t)$ is the occlusion mask with continuous values between 0 and 1. $M$ denotes the mask prediction network. Our occlusion mask is soft instead of binary to better handle the "zoom in" scenario. For example, when an object is moving closer to our camera, the object will become blurrier over time if we only warp previous frames. To increase the resolution of the object, we need to synthesize new texture details. By using a soft mask, we can add details by gradually blending the warped pixels and the newly synthesized pixels.

We use residual networks [26] for $M$, $W$, and $H$. To generate high-resolution videos, we adopt a coarse-to-fine generator design similar to the method of Wang et. al [72].

As using multiple discriminators can mitigate the mode collapse problem during GANs training [19, 66, 72], we also design two types of discriminators as detailed below.

**Conditional image discriminator** $D_I$. The purpose of $D_I$ is to ensure that each output frame resembles a real image given the same source image. This conditional discriminator should output 1 for a true pair $(\mathbf{x}_t, \mathbf{s}_t)$ and 0 for a fake one $(\tilde{\mathbf{x}}_t, \mathbf{s}_t)$.

**Conditional video discriminator** $D_V$. The purpose of $D_V$ is to ensure that consecutive output frames resemble the temporal dynamics of a real video given the same optical flow. While $D_I$ conditions on the source image, $D_V$ conditions on the flow. Let $\mathbf{w}_{t-K}^{t-2}$ be $K-1$ optical flow for the $K$ consecutive real images $\mathbf{x}_{t-K}^{t-1}$. This conditional discriminator $D_V$ should output 1 for a true pair $(\mathbf{x}_{t-K}^{t-1}, \mathbf{w}_{t-K}^{t-2})$ and 0 for a fake one $(\tilde{\mathbf{x}}_{t-K}^{t-1}, \mathbf{w}_{t-K}^{t-2})$.

We introduce two sampling operators to facilitate the discussion. First, let $\phi_I$ be a random image sampling operator such that $\phi_I(\mathbf{x}_1^T, \mathbf{s}_1^T) = (\mathbf{x}_i, \mathbf{s}_i)$ where $i$ is an integer uniformly sampled from 1 to $T$. In other words, $\phi_I$ randomly samples a pair of images from $(\mathbf{x}_1^T, \mathbf{s}_1^T)$. Second, we define $\phi_V$ as a sampling operator that randomly retrieve $K$ consecutive frames. Specifically, $\phi_V(\mathbf{w}_1^{T-1}, \mathbf{x}_1^T, \mathbf{s}_1^T) = (\mathbf{w}_{i-K}^{i-2}, \mathbf{x}_{i-K}^{i-1}, \mathbf{s}_{i-K}^{i-1})$ where $i$ is an integer uniformly sampled from $K+1$ to $T+1$. This operator retrieves $K$ consecutive frames and the corresponding $K-1$ optical flow images. With $\phi_I$ and $\phi_V$, we are ready to present our learning objective function.

**Learning objective function**. We train the sequential video synthesis function $F$ by solving

$$\min_F \left( \max_{D_I} \mathcal{L}_I(F, D_I) + \max_{D_V} \mathcal{L}_V(F, D_V) \right) + \lambda_W \mathcal{L}_W(F), \tag{5}$$

where $\mathcal{L}_I$ is the GAN loss on images defined by the conditional image discriminator $D_I$, $\mathcal{L}_V$ is the GAN loss on $K$ consecutive frames defined by $D_V$, and $\mathcal{L}_W$ is the flow estimation loss. The weight $\lambda_W$ is set to 10 throughout the experiments based on a grid search. In addition to the loss terms in Equation 5, we use the discriminator feature matching loss [40, 72] and VGG feature matching loss [16, 34, 72] as they improve the convergence speed and training stability [72]. Please see the supplementary material for more details.

We further define the image-conditional GAN loss $\mathcal{L}_I$ [33] using the operator $\phi_I$

$$E_{\phi_I(\mathbf{x}_1^T, \mathbf{s}_1^T)}[\log D_I(\mathbf{x}_i, \mathbf{s}_i)] + E_{\phi_I(\tilde{\mathbf{x}}_1^T, \mathbf{s}_1^T)}[\log(1 - D_I(\tilde{\mathbf{x}}_i, \mathbf{s}_i))]. \tag{6}$$

Similarly, the video GAN loss $\mathcal{L}_V$ is given by

$$E_{\phi_V(\mathbf{w}_1^{T-1}, \mathbf{x}_1^T, \mathbf{s}_1^T)}[\log D_V(\mathbf{x}_{i-K}^{i-1}, \mathbf{w}_{i-K}^{i-2})] + E_{\phi_V(\mathbf{w}_1^{T-1}, \tilde{\mathbf{x}}_1^T, \mathbf{s}_1^T)}[\log(1 - D_V(\tilde{\mathbf{x}}_{i-K}^{i-1}, \mathbf{w}_{i-K}^{i-2}))]. \tag{7}$$

Recall that we synthesize a video $\tilde{\mathbf{x}}_1^T$ by recursively applying $F$.

The flow loss $\mathcal{L}_W$ includes two terms. The first is the endpoint error between the ground truth and the estimated flow, and the second is the warping loss when the flow warps the previous frame to the next frame. Let $\mathbf{w}_t$ be the ground truth flow from $\mathbf{x}_t$ to $\mathbf{x}_{t+1}$. The flow loss $\mathcal{L}_W$ is given by

$$\mathcal{L}_W = \frac{1}{T-1} \sum_{t=1}^{T-1} \left( \|\tilde{\mathbf{w}}_t - \mathbf{w}_t\|_1 + \|\tilde{\mathbf{w}}_t(\mathbf{x}_t) - \mathbf{x}_{t+1}\|_1 \right). \tag{8}$$

**Foreground-background prior.** When using semantic segmentation masks as the source video, we can divide an image into foreground and background areas based on the semantics. For example, buildings and roads belong to the background, while cars and pedestrians are considered as the foreground. We leverage this strong foreground-background prior in the generator design to further improve the synthesis performance of the proposed model.

In particular, we decompose the image hallucination network $H$ into a foreground model $\tilde{\mathbf{h}}_{F,t} = H_F(\mathbf{s}_{t-L}^t)$ and a background model $\tilde{\mathbf{h}}_{B,t} = H_B(\tilde{\mathbf{x}}_{t-L}^{t-1}, \mathbf{s}_{t-L}^t)$. We note that background motion can be modeled as a global transformation in general, where optical flow can be estimated quite accurately. As a result, the background region can be generated accurately via warping, and the background hallucination network $H_B$ only needs to synthesize the occluded areas. On the other hand, a foreground object often has a large motion and only occupies a small portion of the image, which makes optical flow estimation difficult. The network $H_F$ has to synthesize most of the foreground content from scratch. With this foreground–background prior, $F$ is then given by

$$F(\tilde{\mathbf{x}}_{t-L}^{t-1}, \mathbf{s}_{t-L}^t) = (\mathbf{1} - \tilde{\mathbf{m}}_t) \odot \tilde{\mathbf{w}}_{t-1}(\tilde{\mathbf{x}}_{t-1}) + \tilde{\mathbf{m}}_t \odot \left( (\mathbf{1} - \mathbf{m}_{B,t}) \odot \tilde{\mathbf{h}}_{F,t} + \mathbf{m}_{B,t} \odot \tilde{\mathbf{h}}_{B,t} \right), \quad (9)$$

where $\mathbf{m}_{B,t}$ is the background mask derived from the ground truth segmentation mask $\mathbf{s}_t$. This prior improves the visual quality by a large margin with the cost of minor flickering artifacts. In Table 2, our user study shows that most people prefer the results with foreground–background modeling. A qualitative comparison is also included in the supplementary material.

**Multimodal synthesis.** The synthesis network $F$ is a unimodal mapping function. Given an input source video, it can only generate one output video. To achieve multimodal synthesis [19, 72, 82], we adopt a feature embedding scheme [72] for the source video that consists of instance-level semantic segmentation masks. Specifically, at training time, we train an image encoder $E$ to encode the ground truth real image $\mathbf{x}_t$ into a $d$-dimensional feature map ($d = 3$ in our experiments). We then apply an instance-wise average pooling to the map so that all the pixels within the same object share the same feature vectors. We then feed both the instance-wise averaged feature map $\mathbf{z}_t$ and the input semantic segmentation mask $\mathbf{s}_t$ to the generator $F$. Once training is done, we fit a mixture of Gaussian distribution to the feature vectors that belong to the same object class. At test time, we sample a feature vector for each object instance using the estimated distribution of that object class. Given different feature vectors, the generator $F$ can synthesize videos with different visual appearances.

## 4 Experiments

**Implementation details.** We train our network in a spatio-temporally progressive manner. In particular, we start with generating low-resolution videos with few frames, and all the way up to generating full resolution videos with 30 (or more) frames. Our coarse-to-fine generator consists of three scales: $512 \times 256$, $1024 \times 512$, and $2048 \times 1024$ resolutions, respectively. The mask prediction network $M$ and flow prediction network $W$ share all the weights except for the output layer. We use the multi-scale PatchGAN discriminator architecture [33, 72] for the image discriminator $D_I$. In addition to multi-scale in the spatial resolution, our multi-scale video discriminator $D_V$ also looks at different frame rates of the video to ensure both short-term and long-term consistency. See the supplementary material for more details.

We train our model for 40 epochs using the ADAM optimizer [39] with lr $= 0.0002$ and $(\beta_1, \beta_2) = (0.5, 0.999)$ on an NVIDIA DGX1 machine. We use the LSGAN loss [49]. Due to the high image resolution, even with one short video per batch, we have to use all the GPUs in DGX1 (8 V100 GPUs, each with 16GB memory) for training. We distribute the generator computation task to 4 GPUs and the discriminator task to the other 4 GPUs. Training takes $\sim 10$ days for 2K resolution.

**Datasets.** We evaluate the proposed approach on several datasets.

Table 1: Comparison between competing video-to-video synthesis approaches on Cityscapes.

| Fréchet Inception Dist. | I3D | ResNeXt | Human Preference Score | short seq. | long seq. |
|---|---|---|---|---|---|
| `pix2pixHD` | 5.57 | 0.18 | `vid2vid` (ours) / `pix2pixHD` | **0.87** / 0.13 | **0.83** / 0.17 |
| `COVST` | 5.55 | 0.18 | `vid2vid` (ours) / `COVST` | **0.84** / 0.16 | **0.80** / 0.20 |
| `vid2vid` (ours) | **4.66** | **0.15** | | | |

Table 2: Ablation study. We compare the proposed approach to its three variants.

| Human Preference Score | |
|---|---|
| `vid2vid` (ours) / `no background-foreground prior` | **0.80** / 0.20 |
| `vid2vid` (ours) / `no conditional video discriminator` | **0.84** / 0.16 |
| `vid2vid` (ours) / `no flow warping` | **0.67** / 0.33 |

Table 3: Comparison between future video prediction methods on Cityscapes.

| Fréchet Inception Dist. | I3D | ResNeXt | Human Preference Score | |
|---|---|---|---|---|
| PredNet | 11.18 | 0.59 | `vid2vid` (ours) / PredNet | **0.92** / 0.08 |
| MCNet | 10.00 | 0.43 | `vid2vid` (ours) / MCNet | **0.98** / 0.02 |
| `vid2vid` (ours) | **3.44** | **0.18** | | |

- **Cityscapes** [13]. The dataset consists of $2048 \times 1024$ street scene videos captured in several German cities. Only a subset of images in the videos contains ground truth semantic segmentation masks. To obtain the input source videos, we use those images to train a DeepLabV3 semantic segmentation network [11] and apply the trained network to segment all the videos. We use the optical flow extracted by FlowNet2 [32] as the ground truth flow $\mathbf{w}$. We treat the instance segmentation masks computed by the Mask R-CNN [25] as our instance-level ground truth. In summary, the training set contains 2975 videos, each with 30 frames. The validation set consists of 500 videos, each with 30 frames. Finally, we test our method on three long sequences from the Cityscapes demo videos, with 600, 1100, and 1200 frames, respectively. We will show that although trained on short videos, our model can synthesize long videos.

- **Apolloscape** [29] consists of 73 street scene videos captured in Beijing, whose video lengths vary from 100 to 1000 frames. Similar to Cityscapes, Apolloscape is constructed for the image/video semantic segmentation task. But we use it for synthesizing videos using the semantic segmentation mask. We split the dataset into half for training and validation.

- **Face video dataset** [57]. We use the real videos in the FaceForensics dataset, which contains 854 videos of news briefing from different reporters. We use this dataset for the sketch video to face video synthesis task. To extract a sequence of sketches from a video, we first apply a face alignment algorithm [38] to localize facial landmarks in each frame. The facial landmarks are then connected to create the face sketch. For background, we extract Canny edges outside the face regions. We split the dataset into 704 videos for training and 150 videos for validation.

- **Dance video dataset.** We download YouTube dance videos for the pose to human motion synthesis task. Each video is about $3 \sim 4$ minutes long at $1280 \times 720$ resolution, and we crop the central $512 \times 720$ regions. We extract human poses with DensePose [23] and OpenPose [7], and directly concatenate the results together. Each training set includes a dance video from a single dancer, while the test set contains videos of other dance motions or from other dancers.

**Baselines.** We compare our approach to two baselines trained on the same data.

- `pix2pixHD` [72] is the state-of-the-art image-to-image translation approach. When applying the approach to the video-to-video synthesis task, we process input videos frame-by-frame.

- `COVST` is built on the coherent video style transfer [10] by replacing the stylization network with `pix2pixHD`. The key idea in `COVST` is to warp high-level deep features using optical flow for achieving temporally coherent outputs. No additional adversarial training is applied. We feed in ground truth optical flow to `COVST`, which is impractical for real applications. In contrast, our model estimates optical flow from source videos.

**Evaluation metrics.** We use both subjective and objective metrics for evaluation.

- **Human preference score.** We perform a human subjective test for evaluating the visual quality of synthesized videos. We use the Amazon Mechanical Turk (AMT) platform. During each

Figure 2: Apolloscape results. Left: `pix2pixHD`. Center: `COVST`. Right: proposed. The input semantic segmentation mask video is shown in the left video. *The figure is best viewed with Acrobat Reader. Click the image to play the video clip.*

Figure 3: Example multi-modal video synthesis results. These synthesized videos contain different road surfaces. *The figure is best viewed with Acrobat Reader. Click the image to play the video clip.*

Figure 4: Example results of changing input semantic segmentation masks to generate diverse videos. Left: tree→building. Right: building→tree. The original video is shown in Figure 3. *The figure is best viewed with Acrobat Reader. Click the image to play the video clip.*

test, an AMT participant is first shown two videos at a time (results synthesized by two different algorithms) and then asked which one looks more like a video captured by a real camera. We specifically ask the worker to check for both temporal coherence and image quality. A worker must have a life-time task approval rate greater than 98% to participate in the evaluation. For each question, we gather answers from 10 different workers. We evaluate the algorithm by the ratio that the algorithm outputs are preferred.

- **Fréchet Inception Distance (FID)** [27] is a widely used metric for implicit generative models, as it correlates well with the visual quality of generated samples. The FID was originally developed for evaluating image generation. We propose a variant for video evaluation, which measures both visual quality and temporal consistency. Specifically, we use a pre-trained video recognition CNN as a feature extractor after removing the last few layers from the network. This feature extractor will be our "inception" network. For each video, we extract a spatio-temporal feature map with this CNN. We then compute the mean $\tilde{\mu}$ and covariance matrix $\tilde{\Sigma}$ for the feature vectors from all the synthesized videos. We also calculate the same quantities $\mu$ and $\Sigma$ for the ground truth videos. The FID is then calculated as $\|\mu - \tilde{\mu}\|^2 + \mathrm{Tr}\left(\Sigma + \tilde{\Sigma} - 2\sqrt{\Sigma\tilde{\Sigma}}\right)$. We use two different pre-trained video recognition CNNs in our evaluation: I3D [8] and ResNeXt [75].

**Main results.** We compare the proposed approach to the baselines on the Cityscapes benchmark, where we apply the learned models to synthesize 500 short video clips in the validation set. As shown in Table 1, our results have a smaller FID and are often favored by the human subjects. We also report the human preference scores on the three long test videos. Again, the videos rendered by our approach are considered more realistic by the human subjects. The human preference scores for the Apolloscape dataset are given in the supplementary material.

Figure 5: Example face→sketch→face results. Each set shows the original video, the extracted edges, and our synthesized video. *The figure is best viewed with Acrobat Reader. Click the image to play the video clip.*

Figure 6: Example dance→pose→dance results. Each set shows the original dancer, the extracted poses, and the synthesized video. *The figure is best viewed with Acrobat Reader. Click the image to play the video clip.*

Figures 1 and 2 show the video synthesis results. Although each frame rendered by `pix2pixHD` is photorealistic, the resulting video lacks temporal coherence. The road lane markings and building appearances are inconsistent across frames. While improving upon `pix2pixHD`, `COVST` still suffers from temporal inconsistency. On the contrary, our approach produces a high-resolution, photorealistic, temporally consistent video output. We can generate 30-second long videos, showing that our approach synthesizes convincing videos with longer lengths.

We conduct an ablation study to analyze several design choices of our method. Specifically, we create three variants. In one variant, we do not use the foreground-background prior, which is termed `no background-foreground prior`. That is, instead of using Equation 9, we use Equation 4. The second variant is `no conditional video discriminator` where we do not use the video discriminator $D_V$ for training. In the last variant, we remove the optical flow prediction network $W$ and the mask prediction network $M$ from the generator $F$ in Equation 4 and only use $H$ for synthesis. This variant is referred to as `no flow warping`. We use the human preference score on Cityscapes for this ablation study. Table 2 shows that the visual quality of output videos degrades significantly without the ablated components. To evaluate the effectiveness of different components in our network, we also experimented with directly using ground truth flows instead of estimated flows by our network. An example can be found in the supplementary material. We found the results visually similar, which suggests that our network is robust to the errors in the estimated flows.

**Multimodal results.** Figure 3 shows example multimodal synthesis results. In this example, we keep the sampled feature vectors of all the object instances in the video the same except for the road instance. The figure shows temporally smooth videos with different road appearances.

**Semantic manipulation.** Our approach also allows the user to manipulate the semantics of source videos. In Figure 4, we show an example of changing the semantic labels. In the left video, we replace all trees with buildings in the original segmentation masks and synthesize a new video. On the right, we show the result of replacing buildings with trees.

**Sketch-to-video synthesis for face swapping.** We train a sketch-to-face synthesis video model using the real face videos in the FaceForensics dataset [57]. As shown in Figure 5, our model can convert sequences of sketches to photorealistic output videos. This model can be used to change the facial appearance of the original face videos [5].

**Pose-to-video synthesis for human motion transfer.** We also apply our method to the task of converting sequences of human poses to photorealistic output videos. We note that the image counterpart was studied in recent works [4, 17, 47, 48]. As shown in Figure 6, our model learns to synthesize high-resolution photorealistic output dance videos that contain unseen body shapes and

Figure 7: Future video prediction results. Top left: ground truth. Top right: PredNet [45]. Bottom left: MCNet [67]. Bottom right: ours. *The figure is best viewed with Acrobat Reader. Click the image to play the video clip.*

motions. Our method can change the clothing [78, 80] for the same dancer (Figure 6 left) as well as transfer the visual appearance to new dancers (Figure 6 right) as explored in concurrent work [1,9,77].

**Future video prediction.** We show an extension of our approach to the future video prediction task: learning to predict the future video given a few observed frames. We decompose the task into two sub-tasks: 1) synthesizing future semantic segmentation masks using the observed frames, and 2) converting the synthesized segmentation masks into videos. In practice, after extracting the segmentation masks from the observed frames, we train a generator to predict future semantic masks. We then use the proposed video-to-video synthesis approach to convert the predicted segmentation masks to a future video.

We conduct both quantitative and qualitative evaluations with comparisons to two start-of-the-art approaches: PredNet [45] and MCNet [67]. We follow the prior work [41,69] and report the human preference score. We also include the FID scores. As shown in Table 3, our model produces smaller FIDs, and the human subjects favor our resulting videos. In Figure 7, we visualize the future video synthesis results. While the image quality of the results from the competing algorithms degrades significantly over time, ours remains consistent.

## 5   Discussion

We present a general video-to-video synthesis framework based on conditional GANs. Through carefully-designed generators and discriminators as well as a spatio-temporal adversarial objective, we can synthesize high-resolution, photorealistic, and temporally consistent videos. Extensive experiments demonstrate that our results are significantly better than the results by state-of-the-art methods. Our method also compares favorably against the competing video prediction methods.

Although our approach outperforms previous methods, our model still fails in a couple of situations. For example, our model struggles in synthesizing turning cars due to insufficient information in label maps. This could be potentially addressed by adding additional 3D cues, such as depth maps. Furthermore, our model still can not guarantee that an object has a consistent appearance across the whole video. Occasionally, a car may change its color gradually. This issue might be alleviated if object tracking information is used to enforce that the same object shares the same appearance throughout the entire video. Finally, when we perform semantic manipulations such as turning trees into buildings, visible artifacts occasionally appear as building and trees have different label shapes. This might be resolved if we train our model with coarser semantic labels, as the trained model would be less sensitive to label shapes.

**Acknowledgements** We thank Karan Sapra, Fitsum Reda, and Matthieu Le for generating the segmentation maps for us. We also thank Lisa Rhee and Miss Ketsuki for allowing us to use their dance videos for training. We thank William S. Peebles for proofreading the paper.

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
