[Supplementary Material]

# Supplementary Material for Video-to-Video Synthesis

**Ting-Chun Wang[1], Ming-Yu Liu[1], Jun-Yan Zhu[2], Guilin Liu[1],**
**Andrew Tao[1], Jan Kautz[1], Bryan Catanzaro[1]**
[1]NVIDIA, [2]MIT CSAIL
{tingchunw,mingyul,guilinl,atao,jkautz,bcatanzaro}@nvidia.com,
junyanz@mit.edu

## 1 Network Architecture

**Generators.** Our network adopts a coarse-to-fine architecture. For the lowest resolution, the network takes in a number of semantic label maps $\mathbf{s}_{t-L}^t$ and previously generated frames $\tilde{\mathbf{x}}_{t-L}^{t-1}$ as input. The label maps are concatenated together and undergo a number of residual blocks to form intermediate high-level features. Same with the previously generated images. Then, these two intermediate layers are added, and fed into two separate residual networks to output the hallucinated image $\tilde{\mathbf{h}}_t$, and the flow map $\tilde{\mathbf{w}}_t$ as well as the mask $\tilde{\mathbf{m}}_t$ (Figure 1).

Next, to build from low-res results to higher-res results, we apply another network $G_2$ on top of the low-res network $G_1$ (Figure 2). Specifically, the inputs are first downsampled and fed into $G_1$. Then, we extract features from the last feature layer of $G_1$, and add it to the feature layer of $G_2$. These summed features are then fed into another series of residual blocks to output the higher resolution images.

**Discriminators.** For discriminators, we adopt the multi-scale patch GAN architecture. To perform our temporally multi-scale approach, we subsample the actual/generated sequences by different amounts to generate different inputs to the temporal discriminators. In the finest scale, we take $K$ consecutive frames in the original sequence as input. In the next scale, we subsample the video by a factor of $K$ (i.e., skipping every $K-1$ intermediate frames), then take consecutive $K$ frames in this new sequence as input. We do this for up to 3 scales in our implementation, and found that this helps us ensure both short-term and long-term consistency.

## 2 Evaluation for the Apolloscape Dataset

We provide both the FID and the human preference score on the Apolloscape dataset. For both metrics, our method outperforms the other baselines.

Table 1: Comparison between competing video-to-video synthesis approaches on Apolloscape.

| Fréchet Inception Dist. | I3D | ResNeXt | Human Preference Score | |
|---|---|---|---|---|
| pix2pixHD | 2.33 | 0.128 | proposed / pix2pixHD | **0.61** / 0.39 |
| COVST | 2.36 | 0.128 | proposed / COVST | **0.59** / 0.41 |
| proposed | **2.24** | **0.125** | | |

## 3 Example Results

**Long sequence result.** We provide an example 30 second video sequence in 2k resolution in the supplementary material. Please see stuttgart_01_synthesized.mp4.

Figure 1: The network architecture ($G_1$) for low-res videos. Our network takes in a number of semantic label maps and previously generated images, and outputs the intermediate frame and the flow map along with the mask.

Figure 2: The network architecture ($G_2$) for higher resolution videos. The label maps and previous frames are downsampled and fed into the low-res network $G_1$. Then, the features from the high-res network and the last layer of the low-res network are summed and fed into another series of residual blocks to output the final images.

**Semantic manipulation result.** In addition, we provide an example of semantically manipulated video. The input segmentation mask is changed from time to time during the sequence, from trees to buildings or vice versa. The styles of the buildings, roads, etc, are also changed within the video. Please see `stuttgart_01_manipulated.mp4`.

**Result using ground truth flows.** We did an experiment where we replaced the estimated flows with the ground truth flows during inference while keeping the rest of the model the same. We found the new outputs remain visually similar to the original ones, as our flow estimation is reasonably accurate with an average end-point-error (AEPE) of $2.14$. Note that the state-of-the-art AEPE on KITTI 2012 is $1.7$, and our flow estimation is much harder as it is based on segmentation masks and past generated frames, rather than real video sequences. This suggests that the estimated flows, albeit not perfect, are sufficient for our task. Please see `stuttgart_01_gtFlow.mp4` for results using ground truth flow.

**Result without using the foreground-background prior.** We provide a qualitative example where the foreground-background prior is not adopted in the synthesis network. We found that the visual quality drops by a large margin, which correlates well with the subjective study performed in the main paper (Table 2). Please see `stuttgart_01_noFG.mp4`.