[Reviews · NeurIPS 2018]

Reviewer 1



This paper focuses on video-2-video synthesis, i.e. given a real video the goal is to learn a model that outputs a new photorealistic and temporally consistent video with (ideally) the same data distribution, preserving the content and style of the source video. Existing image-2-image methods produce photorealistic images, but they do not account for the temporal dimension, resulting in high-frequency artifacts across time. This work builds on existing image-2-image works and mainly extends them into the temporal dimension to ensure temporal coherence. By employing conditional GANs the method provides high-level control over the output, e.g. certain parts of the dynamic scene can be introduced or replaced though segmentation masks. Although the theoretical background and components are employed from past work, there is significant amount of effort in putting them together and adding the temporal extension. Although this work does not solve the problem per se, it seems like an important step for which the community would be interested in, while the authors also explicitly promise to release their implementation. The results look reasonable, though I would like to see a more involved discussion about current disadvantages and future directions. Furthermore, some clarifications are needed and suggestions for further experiments are given. Although I am not an expert in this narrow field, I am leaning towards supporting this work, on the condition that feedback will be incorporated. More detailed feedback follows below. - L10, L38 - I am not very convinced about the novelty of the 'new generator and discriminator architectures', in the technical section they are discussed as straightforward extensions. - L167-168 - My understanding is that this is not what you do. So I am not sure about the use of this comment in the manuscript. Can you please comment further? What is used for the provided results in the manuscript and supplementary material? The suggested speedup trick would force hallucination for the foreground areas, which is in contrast with the goal of this work as explicitly mentioned in L3 (and elsewhere), temporal consistency and content preservation would be harder for the foreground. If, in contrast to my understanding, this is what you do, can you please be more specific about the trade-offs? - L119 - The phrase 'between 0 and 1' sounds a bit ambiguous, the values are binary. - Eq4 - Maybe you can help the reader with more intuition, e.g. the first part corresponds to non-occluded regions (synthesising by warping) while the second corresponds to occluded regions (synthesising by hallucinating). - Eq7 - In the first term, shouldn't the superscript for w be i-2 ? - Eq8, L153 - It is counter-intuitive to see the two lambdas to have the same value. Maybe you can combine them in one lambda and move it from Eq8 to Eq5 ? - The lambda(s) in Eq8(Eq5) is an important steering weight for the training objective, how do you set this? I guess it is too expensive to provide evaluation, so some intuition would be helpful. - L159 - Please compare with L160, it seems there is a \widetilde{x} (etc) missing in H_F. - L192, L193 - I would love to hear some more specific details in the supplementary material, for better self-consistency. Why is this mentioned here and not after Eq5? - Nice idea to include videos in the pdf. There is a clear improvement in the qualitative results, however I would like to hear some discussion about current disadvantages and pointers for future directions. In the long videos of the supplementary material replacing objects through the segmentation masks is interesting but has obvious artifacts (buildings, trees), any pointers for improvements? - L108 - Can you please discuss how you set L=2? It would be great if you could provide at least qualitative comparison for a short video that gives intuition to the reader. Furthermore, could you please add a comment on what is the price to pay for a bigger L in terms of modelling/training difficulty? - The method employs optical flow estimation and occlusion mask estimation (L123), and both are sources of noise. It would be great if you could replace estimation of them with ground truth on some synthetic dataset, this could give pointers and prioritisation on what would improve your results (e.g. parts of lanes disappear/blur after some frames). - minor typos: - L120 - 'of the image' - L274 - 'allows the user' - L275 - 'synthesise a new' - Thank you for explicitly promising to release code/models, reproducibility could be hard without this. ================================== Updates after authors' response: - I welcome the authors' responce and clarifications. My new comments are again towards constructive refining of the manuscript. Some additions are asked below, in order to incorporate the responce in the manuscript in case of acceptance. I understand that there are space limitations, so the authors should do their best to communicate the points in the main paper even more briefly than in the rebuttal, but worst case they should be in the supplementary material and video. - "Using ground truth flows" - Thank you for performing the requested experiment. Please incorporate in the paper and in the supplementary video. - "Limitations and future work" - Thank you for the discussion. I often feel that in certain communities this is underplayed, while it points the community to interesting steps. Again, please incorporate in the paper and in the supplementary video. For point 3 I am not fully convinced about the coarser semantic labels (training less affected by the label shapes), but maybe I am missing something. - Thank you for offering to downplay the "new" term and novelty of architecture. I do agree that the extension is not trivial. I am not sure about the comment on combining different resolution, if I remember correctly this is not tackled. - "Synthesis in the foreground area" - Please add in the paper the comment about flickering and "visual quality". I am not sure what you mean with the term "visual quality", so please reword to communicate it better. The flag in the code is a great idea. Please add a qualitative comparison in the video, it would be very informative wrt the trade-off between image quality and decision on FG synthesis. - "Setting parameters and losses" - Please add very brief comment on similarity for 1..10. - L=2 is chosen but L=4 (more temporal info/frames) gives worse performance. This shows that the temporal extension is not trivial, and points that more future work is needed to cleverly use temporal information. This should be part of "future work" and must be mentioned. This is not to downplay this work, just to underline that the problem is non-trivial. Please mind that the resolution due to time constraints was decreased for this experiment.

Reviewer 2



This paper proposes a method for video-to-video synthesis (as the title states). It is like Image-to-image translation but for videos. In this way they are able to generate HD images (2MP) with highly temporal coherence and very photorealistic. The method tries to model the conditional distribution using a GAN. Many GAN improvements are present such as several discriminators, multi-scale patch discriminator, or coarse-to-fine approach. Also it makes use of Optical flow. The method is compared both qualitatively and quantitatively with state of the art. It shows a clear improve on all the experiments. It has been tested in driving videos and in faces. Two extra experiments are very nice: modifying classes and predicting next frames. Minnor comments: - Remove text from the videos Comments after rebuttal ---------------------------------- The authors have addressed most of the points of the reviewers. The answer for using optical flow groundtruth instead of the computed one is surprising. Usually optical flow estimation results are not very good. However the method is robust to these errors. Maybe the model is not relaying much on the optical flow? The analysis of the failure cases is very interesting.

Reviewer 3



------- Summary --------: The paper addresses the video-to-video synthesis problem. It considers translating a video from a source domain to a target domain, which the target domain is high-resolution, realistic videos. The source domain on 2 of the datasets (Cityscape, Apolloscape) is semantic segmentation map and in the other one (Face Video dataset) is face sketch (obtained by face landmarks). In addition they generate frames of the future of the video. Their approach is a spatial-temporal adversarial learning framework. They designed an objective function, consisting 1) a sequential generator, 2) conditional video generator (help estimating optical flow), and 3) conditional image generator. They also employ segmentation labels to identify background/foreground information and leverage this information to improve the temporal consistency, and quality of the foreground objects (i.e: pedestrians, cars), and having a better motion estimation. Moreover, they present the results for longer term video prediction conditioning on the source domain image, and former generated frames. In addition, they present a sample of image manipulation. --------------------------------------------------------------------------------------- I believe that the paper is valuable to the NIPS research community. It targets a new and challenging problem of video-to-video translation. The paper has good quality and clearity. There are some strong claims in the paper might not be accurate, and the related work part can improve: Conditional generation in videos is highlighted as one of the main contributions of the paper in several parts of the paper. In the related work section, introduction and further part of the paper, it is claimed that this paper is the first paper on conditional video generation using GAN. However, the concept of conditioning in video generation is not a novel idea. Bellow, I put a few references on video prediction using conditional generation: Generating Videos with Scene Dynamics, Vondrick et al. Stochastic Adversarial Video Prediction, Lee et al. Unsupervised Learning of Disentangled Representations from Video, Denton et al. Video Pixel Networks is a conditional model. Spatio-Temporal consistency: The paper claims due to their objective function and the design of the adversarial training, they have spatio-temporal consistency. Videos in the supplementary materials, and the quantitative evaluations in the paper support the claim but there is not much literature review for other approaches addressing the same issue. Basically spatio-temporal consistency is the main problem addressed in most of the video prediction papers. Please discuss it in the literature review. Video generation by estimating/employing Optical Flow, was used in the literature. The format and the formulation was different with what was presented in the paper. However, it is good to cite the relevant work on this area. For example: *Hierarchical Video Generation from Orthogonal Information: Optical Flow and Texture*, Ohnishi et al. ----------------------------------------